# Understanding the current acute aortic syndrome (AAS) pathways—The Collaborative Acute Aortic Syndrome Project (CAASP) protocol

**Jim Zhong**[1,2,3]*, **Ganesh Vigneswaran**[3,4,5], **Nawaz Z. Safdar**[2,3,6], **Indrajeet Mandal**[3,7], **Aminder A. Singh**[8,9], **Sandip Nandhra**[8,10], **on behalf of the CAASP steering committee**[¶]

1 Department of Diagnostic and Interventional Radiology, Leeds Teaching Hospitals NHS Trust, Leeds, United Kingdom, 2 School of Medicine, University of Leeds, Leeds, United Kingdom, 3 United Kingdom Interventional Radiology Trainee Research (UNITE) Collaborative, London, United Kingdom, 4 Faculty of Medicine, School of Cancer Sciences, University of Southampton, Southampton, United Kingdom, 5 Department of Diagnostic and Interventional Radiology, University Hospital Southampton NHS Foundation Trust, Southampton, United Kingdom, 6 Department of Internal Medicine, Leeds Teaching Hospitals NHS Trust, Leeds, United Kingdom, 7 Department of Radiology, Oxford University Hospitals, Oxford, United Kingdom, 8 Vascular and Endovascular Research Network (VERN), London, United Kingdom, 9 Cambridge Vascular Unit, Cambridge University Hospitals NHS Foundation Trust, Cambridge, United Kingdom, 10 Department of Vascular Surgery, Newcastle upon Tyne Hospitals, Newcastle, United Kingdom

¶ Membership of the CAASP steering committee is provided in the Acknowledgments Section
* jim.zhong@nhs.net

## Abstract

### Background

Acute aortic syndrome (AAS) is an emergency associated with high peri-hospital mortality rates. Variable clinical presentation makes timely diagnosis challenging and such delays in diagnosis directly impact patient outcomes.

### Aims and objectives

The aims of the Collaborative Acute Aortic Syndrome Project (CAASP) are to characterise and evaluate the current AAS pathways of a cohort of hospitals in the UK, USA and New Zealand to determine if patient outcomes are influenced by the AAS pathway (time to hospital admission, diagnosis and management plan) and demographic, social, geographic and patient-specific factors (clinical presentation and comorbidities). The objectives are to describe different AAS pathways and time duration between hospital admission to diagnosis and management plan instigation, and to compare patient outcomes between pathways.

### Methods

The study is a multicentre, retrospective service evaluation project of adult patients diagnosed on imaging with AAS. It will be coordinated by the UK National Interventional Radiology Trainee Research (UNITE) network and Vascular and Endovascular Research Network (VERN) in conjunction with The Aortic Dissection Charitable Trust (TADCT). All AAS cases

**Data Availability Statement:** No datasets were generated or analysed during the current study. All

relevant data from this study will be made available upon study completion.

**Funding:** The study was supported by the Aortic Dissection Charitable Trust (TADCT).

**Competing interests:** The authors have declared that no competing interests exist.

diagnosed on imaging between 1st January 2018 to 1st June 2021 will be included and followed-up for 6 months. Eligibility criteria include aortic dissection (AD) Type A, Type B, non A/B, penetrating aortic ulcer, and intramural haematoma. Exclusion criteria are non-AAS pathology, acute on chronic AAS, and age<18. This project will evaluate patient demographics, timing of presentation, patient symptoms, risk factors for AD, physical examination findings, timing to imaging and treatment, hospital stay, and mortality. Univariate and multivariate analysis will be used to identify predictors associated with prolonged time to diagnosis or treatment and mortality at 30 days.

## Introduction

### Background

Acute aortic syndrome (AAS) includes a heterogenous group of patients with interlinked aortic pathologies, including penetrating atherosclerotic aortic ulcer (PAU), intramural haematoma (IMH), and aortic dissection (AD). AAS is an emergency with an incidence of 6/100,000 per year [1] and results in high peri-hospital mortality rates [2]. AD accounts for over 90% of AAS cases [3]. There are multiple risk factors for AD including those that increase intimal shear stress, such as hypertension, and those which cause weakening of the vessel wall, such as atherosclerosis or connective tissue disorders comprising Marfan, Ehlers- Danlos and Loeys-Dietz syndromes. A number of patients will also have a family history of aortic dissection, with 13–22% of patients affected having a first degree relative with previous AD [4].

There are approximately 2500 cases per year in England although numerous groups have suggested the condition is under-reported [5,6]. Clinical presentation of AD can be varied, with symptoms such as chest and back pain that can mimic a range of other more common conditions. To make a timely diagnosis, a high degree of clinical suspicion alongside a low threshold for cross-sectional imaging is required. Prompt diagnosis is key to the successful management of patients with acute AD, however in 16–40% of cases there is a delay in diagnosis [6].

At present there is limited understanding of the current pathways used to diagnose AAS. Clinical red-flag symptoms, such as truncal pain and syncope, have historically demonstrated limited accuracy [7]. Although literature suggests that one of the more sensitive (62–78% [8]) symptoms for AAS is truncal pain, it is a very common presenting complaint and may represent several other differential diagnoses. Pain characteristics should therefore be explored in greater detail; however, these are still unreliable at ruling out AAS (sensitivity<90%). Furthermore, there are several factors that delay diagnosis such as female sex, age >70, diabetes mellitus, and painless presentation [9].

Currently, D-dimer is a promising point-of-care biomarker available for the testing of AD [10]. Previous literature has demonstrated high sensitivity if used to rule out AD within the first six hours of symptom onset. Despite this, D-dimer lacks specificity since it may be elevated in several other conditions. Transthoracic echocardiography (TEE) and computed tomography angiography (CTA) continue to be the gold standard diagnostic techniques for AAS, however, there are new biomarkers on the horizon [11] that could prove to be specific in diagnosing AAS [12]. Suitable patient selection for imaging still constitutes a major diagnostic dilemma, with concerns for radiation exposure risks in low-suspicion cases and busy radiology departments with finite imaging capacity playing a role in the decision to proceed to cross-sectional imaging.

### Rationale

Significant variation exist in AAS diagnostic pathways across regions with the majority of emergency departments not having a formal work-up pathway for AAS [1]. This likely affects the time to diagnosing AAS, which is time-sensitive, given the need for urgent management and potential treatment. These diagnostic delays could impact patient outcomes. CAASP has been developed to characterise and understand the current patient pathways for those presenting with suspected AAS, and to highlight clinical factors or diagnostic/ management delays that might impact on patient outcomes.

### Objectives

The primary objective of CAASP is to outline the current national diagnostic and management pathways for patients presenting with suspected AAS and assess the impact of this on early mortality, and to inform and facilitate improvements in clinical practice.

Secondary objectives include:

- To assess the variation in positive yield rate of AAS from all imaging referrals performed to investigate for AAS at each centre.

- To identify any clinical and demographic factors associated with delays in admission, diagnosis and treatment

- To compare differences in diagnostic pathways and outcomes between tertiary aortic centres and other hospitals.

- To identify geographical and socioeconomic variation in AAS detection and outcomes.

- To identify the impact of COVID-19 on AAS pathways

- To understand differences between the UK AAS pathways with international sites (USA/ New Zealand).

- Promote collaborative working between multi-disciplinary trainee-led research collaboratives.

## Methods

### Study design

This is a multicentre, retrospective service evaluation project of adult patients diagnosed on imaging with acute aortic syndrome (AAS). The summary diagram for CAASP is shown in Fig 1. The study will be delivered through the Vascular and Endovascular Research Network (VERN), UK National Interventional Radiology Trainee Research (UNITE) network, the British Society of Interventional Radiology Trainees (BSIRT) network, and the Society of Interventional Radiology Resident and Fellow Section Research and Innovation (SIR RFS R&I) Committee (a collaboration with Massachusetts General Hospital, irlab.mgh.harvard.edu). VERN, UNITE, and SIR RFS R&I group are all trainee-led national research collaboratives that are run by, and engage with, research-active vascular and interventional radiology trainees, and allied health professionals. The co-investigators have experience in designing and delivering multi-centre collaborative studies.

### Eligibility criteria

Patients may be eligible for the study if they meet all inclusion criteria and none of the exclusion criteria apply. Adult patients with a diagnosis of AAS (AD Type A, AD Type B, AD non-

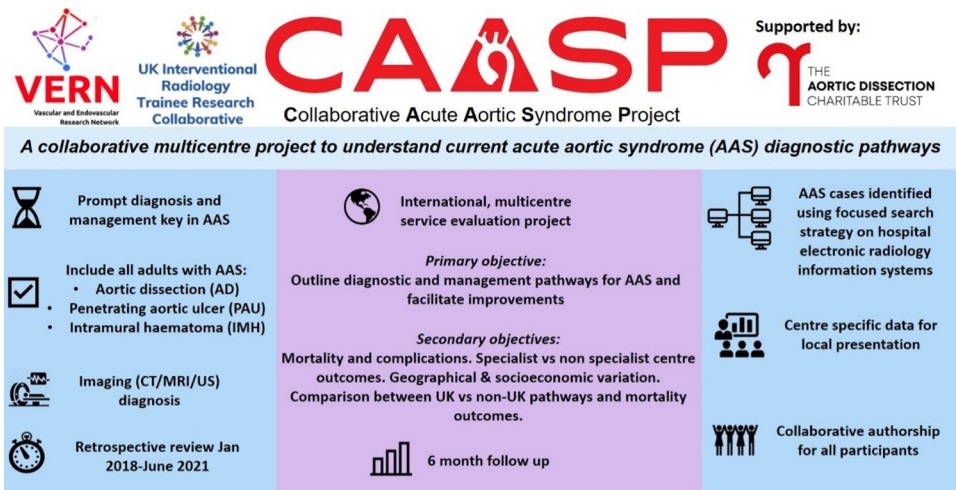

**Fig 1. CAASP summary diagram.**

A non-B, PAU, IMH) on imaging between 1-Jan-2018 and 1-June-2021 will be eligible to be included. Exclusion criteria are non-AAS pathology (e.g. traumatic or iatrogenic aortic injury), chronic AAS presentation and age <18 years.

## Primary outcomes

The primary outcome will be to assess the association of recorded time delays in AAS pathways to admission, imaging diagnosis and treatment planning with patient mortality within 30, 90 days and 6 months

## Secondary outcomes

- The proportion of positive scans from all imaging referrals performed to investigate for AAS and the proportion of AAS cases diagnosed incidentally per site.

- The clinical (presentation symptoms, signs, observations, vital signs, comorbidities and blood test results) and demographic (age, gender, ethnicity, distance to hospital, deprivation score) factors associated with delays in admission, diagnosis and treatment.

- Differences in time durations (to admission, diagnosis and treatment) and mortality outcomes between tertiary aortic centres and other hospitals.

- Compare the impact of geographical (distance to hospital) and socioeconomic (deprivation score) on the AAS diagnosis times and mortality outcomes between sites.

- Evaluate the incidence of AAS and mortality rates before and during COVID-19.

- Compare the UK AAS pathways with USA/ New Zealand pathways and associated time and mortality outcomes.

## Sample size

This project is designed to evaluate current AAS pathways therefore no minimum sample size is required to show effect. Centres with less than 5 cases will be excluded.

## Patient identification

Using a focused search strategy on electronic radiology information systems (S1 File) at each enrolled centre, cases AAS will be identified out of all the CT scans where AAS is queried. Data will be retrospectively collected for cases of AAS diagnosed between 1-Jan-2018 to 1-June-2021. From the first imaging diagnosis of AAS, follow-up data will be collected for a duration of six months. The start date for centres is variable to allow for local registration and approval however the proposed start date is August 2022 and end date March 2023. Prospectively maintained local databases of acute aortic syndrome cases will also be interrogated to ensure identification of all possible cases.

## Data collection

Data collected will include patient demographics, details regarding timing of presentation, symptoms at presentation, risk factors for AD, physical examination findings, investigations including laboratory analyses and imaging, treatment, details regarding hospital stay, and mortality (S1 File). These data will be collected at each hospital site and pseudo-anonymised using a unique study ID which will include the first two letters of the site name and a number. The local site lead will keep a log that contains the key to the study IDs, i.e. a record of the personal identification data linked to each patient study ID number. This record will be filed at the investigational site and only accessed by the local site team. The pseudo-anonymised information will be collected at each participating centre onto a pre-piloted spreadsheet provided by the CAASP study team. The data collection proforma will be circulated to site leads across all centres. Only the pseudo-anonymised data will be transferred to the central study team leads via the secure NHS email server.

## Statistical considerations

Descriptive and summary statistics will be used to present data regarding time to presentation/ time to CT scan/ time to treatment, mortality and complications for AAS patients within the first six months of imaging diagnosis. We will explore significant predictive factors leading to both mortality and delay using univariate and multivariate analysis. Data will be analysed in multiple ways, such as, stepwise multiple linear regression to calculate coefficients of predicted delay times (or delay time ratios) for individual factors that contribute to delays in diagnosis and management as well as multiway ANOVA (analysis of variance). Additionally, we will explore the possibility of utilising different machine learning algorithms such as MLR (multinomial logistic regression), SVM (support vector machines) and Random forests to form predictive models of AAS patient outcomes that later might guide the generation of predictive tools. Sub analysis will also look to focus on differences in diagnostic pathways between specialist cardiovascular centres and non-cardiovascular centres, geographical and socioeconomic (using index of multiple deprivation scores) variation to provide an insight in the variability of management. Statistical significance will be measured at $p < 0.05$ with bonferroni correction to allow for multiple comparisons.

## Ethics and dissemination

**Ethical considerations and confidentiality.** This study will be carried out at participating sites in the UK, USA and New Zealand. CAASP has been approved as a national service evaluation project by the lead organisation Leeds Teaching Hospitals NHS Trust (Approval date 18/05/2022, ref: CAASP). All UK sites participating in the study will be required to register the study as a national service evaluation project with their relevant local departments for approval

prior to data collection. Evidence of this must be shared with the study leads prior to submission of data. Similarly, non-UK centres will be required to show appropriate evidence according to local regulations, for example, institutional review board or Caldicott guardian approval.

Inclusion of patients to the study will not affect clinical decision-making and the standard of care will be upheld. Patients will not be contacted at any point. Therefore, no additional risk will be posed to the study patients. All data will be held anonymised at the point of collection and no patient identifiable information will be stored by the CAASP team. The local site lead will be responsible of keeping one database with local identifiers and an assigned Centre number / patient number (pseudonymised). They will also require a second sheet which only includes the pseudonymised assigned patient/study number. Only the pseudonymised data sheet will be transferred via the secure NHS mail.

**Patient and public involvement.** This project is supported by the TADCT (https://aorticdissectioncharitabletrust.org/), a patient charity, who have been consulted on the rationale for and design of the project. They have contributed towards the choice of data variables to be collected such as the time from admission to imaging diagnosis and clinical presentation symptom characteristics. Data collection will be carried out by clinical staff who are otherwise salaried and will be contributing for free in exchange for appropriate recognition in the research output of this study.

**Dissemination.** Our aim is to evaluate current acute aortic syndrome (AAS) pathways and consequent patient outcomes in a collaborative model. The outcome and data will be used to inform and develop an improved national patient care pathway via The Aortic Dissection Charitable Trust (TADCT). Results of this study will be disseminated via presentations at appropriate scientific meetings and conferences, and publication within relevant peer-reviewed journals. Results will also be shared with members of the TADCT.

**Authorship.** Authorship will involve named individuals involved in the project design (CAASP steering group), and manuscript preparation with the UK IR Trainee Research group and VERN, with individuals collecting data at hospital sites being specifically named as collaborators. Individuals from each centre are estimated to be in the order of six study personnel. These can be medical or allied health professionals with one named lead clinician. Data collectors will be recognised under a collaborative authorship model on subsequent publications.

**Funding.** The Aortic Dissection Charitable Trust will help fund any fees associated with local site project approval and open access publication fees for subsequent manuscripts from this project. There is no other external third-party funding.

## Discussion

CAASP will provide invaluable information regarding the characterisation of AAS diagnostic paths and the identification of critical factors influencing patient outcomes. Delivering the project through a collaborative, pan speciality research model will ensure its success in achieving the required number of cases for statistical analysis. Engaging collaborators will foster connections and bring together a group of clinicians with interest in improving outcomes for AAS patients. This database of collaborators can be used in future studies which can focus on pathway improvement. The co-investigators of CAASP have significant experience of delivering collaborative projects using this model [13–17] and this expertise will ensure project success.

The limitations of this work include its retrospective design. It is felt that first defining current AAS pathways is required, which will allow for future prospective work and inform trial design. There may be geographical bias in the data depending on which centres sign up which could limit generalisability of findings.

## Supporting information

**S1 File.**
(DOCX)

## Acknowledgments

CAASP Steering Committee

Dr Jim Zhong (Lead)–Clinical Research Fellow, University of Leeds/ Interventional Radiology Fellow, Leeds Teaching Hospitals NHS Trust, Leeds, UK

Dr Ganesh Vigneswaran–NIHR Clinical Lecturer in Interventional Radiology, University of Southampton, Southampton, UK

Mr Sandip Nandhra–Consultant Vascular Surgeon, Newcastle upon Tyne Hospitals, Newcastle, UK

Mr Aminder Singh–Vascular Surgery Registrar, Cambridge University Hospitals, Cambridge, UK

Mr Nikesh Dattani–Consultant Vascular & Endovascular Surgeon, Leicester Vascular Institute, Glenfield Hospital, Leicester, UK

Ms Ruth Benson–Academic Clinical Lecturer in Vascular Surgery, University of Birmingham Clinical Trials Unit

Mr Robert Blair–Specialist Registrar in Vascular Surgery, Belfast Health and Social Care Trust

Dr Shian Patel–Consultant Interventional Radiologist, University Hospitals Dorset NHS Trust

Mr Joseph Shalhoub—Consultant Vascular Surgeon, Imperial College Healthcare NHS Trust

Dr Hunain Shiwani—Clinical Research Fellow, University College London, London, UK/ Radiology Registrar, Leeds Teaching Hospitals NHS Trust, Leeds, UK

Mr Nawaz Safdar–Medical Student, University of Leeds, Leeds, UK

Dr Deevia Kotecha–Radiology Registrar, Manchester University NHS Foundation Trust, UK

Dr Avik Som (Lead)—Interventional and Diagnostic Radiology Residency, PGY-4, Massachusetts General Hospital, Boston, USA

Miss Ginny Sun–Medical Student, Harvard University, Boston, USA

Mr Manar Khashram–Vascular Surgeon, Waikato Hospital, Waikato, New Zealand

Mr Graham Cooper–Consultant Cardiac Surgeon, Sheffield Teaching Hospitals, Sheffield, UK / TADCT Research Committee

Professor Julie Sanders—Director Clinical Research, St Bartholomew's Hospital (member of Cardiothoracic Interdisciplinary Research Network)

Dr Philip Scott—Programme Director, Institute of Management & Health, University of Wales Trinity St David

Dr Sarah Wilson—Emergency Medicine Consultant and Deputy Chief of Service, Wexham Park Hospital Emergency Department, Frimley Health NHS Foundation Trust, UK

Dr Robin Williams—Consultant Interventional Radiologist, The Newcastle upon Tyne Hospitals NHS Foundation Trust, Newcastle, UK

Dr Paul Walker—Consultant Interventional Radiologist, Leeds Teaching Hospitals NHS Trust, Leeds, UK

Dr Phil Jackson—Consultant Anaesthetist and ICU Physician, Leeds Teaching Hospitals NHS Trust, Leeds, UK

Dr Mathew Bromley—Consultant Anaesthetist and ICU Physician, Leeds Teaching Hospitals NHS Trust, Leeds, UK

Sources of funding: The Aortic Dissection Charitable Trust (TADCT)

## Author Contributions

**Conceptualization:** Jim Zhong, Aminder A. Singh, Sandip Nandhra.

**Data curation:** Jim Zhong, Ganesh Vigneswaran, Nawaz Z. Safdar, Indrajeet Mandal, Aminder A. Singh, Sandip Nandhra.

**Formal analysis:** Jim Zhong, Ganesh Vigneswaran, Nawaz Z. Safdar, Indrajeet Mandal, Aminder A. Singh, Sandip Nandhra.

**Funding acquisition:** Jim Zhong, Sandip Nandhra.

**Investigation:** Jim Zhong, Ganesh Vigneswaran, Nawaz Z. Safdar, Indrajeet Mandal, Aminder A. Singh, Sandip Nandhra.

**Methodology:** Jim Zhong, Ganesh Vigneswaran, Nawaz Z. Safdar, Indrajeet Mandal, Aminder A. Singh, Sandip Nandhra.

**Project administration:** Jim Zhong, Nawaz Z. Safdar, Indrajeet Mandal, Aminder A. Singh.

**Resources:** Jim Zhong, Ganesh Vigneswaran, Nawaz Z. Safdar, Aminder A. Singh, Sandip Nandhra.

**Software:** Jim Zhong, Ganesh Vigneswaran, Sandip Nandhra.

**Supervision:** Jim Zhong, Sandip Nandhra.

**Validation:** Jim Zhong, Ganesh Vigneswaran, Aminder A. Singh, Sandip Nandhra.

**Visualization:** Jim Zhong, Ganesh Vigneswaran, Sandip Nandhra.

**Writing – original draft:** Jim Zhong, Ganesh Vigneswaran, Nawaz Z. Safdar, Indrajeet Mandal, Aminder A. Singh, Sandip Nandhra.

**Writing – review & editing:** Jim Zhong, Sandip Nandhra.

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
