## [Decision Letter · Decision Letter 0]

13 Oct 2023

PONE-D-23-11061Understanding the current acute aortic syndrome (AAS) pathways - the Collaborative Acute Aortic Syndrome Project (CAASP) protocolPLOS ONE

Dear Dr. Zhong,

Thank you for submitting your manuscript to PLOS ONE. After careful consideration, we feel that it has merit but does not fully meet PLOS ONE’s publication criteria as it currently stands. Therefore, we invite you to submit a revised version of the manuscript that addresses the points raised during the review process.

Please revise.

We look forward to receiving your revised manuscript.

Kind regards,

Academic Editor

PLOS ONE

Journal Requirements:

2. One of the noted authors is a group or consortium [CAASP steering committee ]. In addition to naming the author group, please list the individual authors and affiliations within this group in the acknowledgments section of your manuscript. Please also indicate clearly a lead author for this group along with a contact email address

Reviewers' comments:

Reviewer's Responses to Questions

**Comments to the Author**

1. Does the manuscript provide a valid rationale for the proposed study, with clearly identified and justified research questions?

Reviewer #1: Yes

Reviewer #2: Yes

2. Is the protocol technically sound and planned in a manner that will lead to a meaningful outcome and allow testing the stated hypotheses?

Reviewer #1: Yes

Reviewer #2: Yes

3. Is the methodology feasible and described in sufficient detail to allow the work to be replicable?

Reviewer #1: Yes

Reviewer #2: Yes

4. Have the authors described where all data underlying the findings will be made available when the study is complete?

Reviewer #1: No

Reviewer #2: Yes

5. Is the manuscript presented in an intelligible fashion and written in standard English?

Reviewer #1: Yes

Reviewer #2: Yes

6. Review Comments to the Author

You may also provide optional suggestions and comments to authors that they might find helpful in planning their study.

Reviewer #1: the protocol "Understanding the current acute aortic syndrome (AAS) pathways - the Collaborative

Acute Aortic Syndrome Project (CAASP) idea is good for future review after data collection and analysis.

'

Reviewer #2: The study protocol designed by the authors looks excellent. The delay in diagnosis and treatment are one of the major causes of morbidity and mortality in AAS. I am eagerly waiting for the study results in the future.

I would suggest, that the authors add a protocol diagram / algorithm / flow chat, for a better understanding of the readers.

7. PLOS authors have the option to publish the peer review history of their article (what does this mean?). If published, this will include your full peer review and any attached files.

Reviewer #1: No

Reviewer #2: **Yes: **Mohammed Idhrees

---

## [Author Response · Author response to Decision Letter 0]

16 Nov 2023

Dear Dr Robert Jeenchen Chen and reviewers,

Thank you for your review of our article entitled: Understanding the current acute aortic syndrome (AAS) pathways - the Collaborative Acute Aortic Syndrome Project (CAASP) protocol.

We have provided a point by point response to the reviewer comments in bold.

RESPONSE: We have followed the PLOS ONE guidance and edited the manuscript.

2. One of the noted authors is a group or consortium [CAASP steering committee ]. In addition to naming the author group, please list the individual authors and affiliations within this group in the acknowledgments section of your manuscript. Please also indicate clearly a lead author for this group along with a contact email address

RESPONSE: This has been added in the acknowledgements section at the start of the manuscript.

RESPONSE: References have been reviewed. No retracted papers were cited.

Reviewers' comments:

Reviewer's Responses to Questions

Comments to the Author

1. Does the manuscript provide a valid rationale for the proposed study, with clearly identified and justified research questions?

Reviewer #1: Yes

Reviewer #2: Yes

2. Is the protocol technically sound and planned in a manner that will lead to a meaningful outcome and allow testing the stated hypotheses?

Reviewer #1: Yes

Reviewer #2: Yes

3. Is the methodology feasible and described in sufficient detail to allow the work to be replicable?

Reviewer #1: Yes

Reviewer #2: Yes

4. Have the authors described where all data underlying the findings will be made available when the study is complete?

Reviewer #1: No

Reviewer #2: Yes

 5. Is the manuscript presented in an intelligible fashion and written in standard English?

Reviewer #1: Yes

Reviewer #2: Yes

6. Review Comments to the Author

You may also provide optional suggestions and comments to authors that they might find helpful in planning their study.

Reviewer #1: the protocol "Understanding the current acute aortic syndrome (AAS) pathways - the Collaborative Acute Aortic Syndrome Project (CAASP) idea is good for future review after data collection and analysis.'

Reviewer #2: The study protocol designed by the authors looks excellent. The delay in diagnosis and treatment are one of the major causes of morbidity and mortality in AAS. I am eagerly waiting for the study results in the future.

I would suggest, that the authors add a protocol diagram / algorithm / flow chat, for a better understanding of the readers.

RESPONSE: This has been added. Please see the CAASP Summary Diagram (Figure 1)

7. PLOS authors have the option to publish the peer review history of their article (what does this mean?). If published, this will include your full peer review and any attached files.

Do you want your identity to be public for this peer review? For information about this choice, including consent withdrawal, please see our Privacy Policy.

Reviewer #1: No

Reviewer #2: Yes: Mohammed Idhrees

---

## [Decision Letter · Decision Letter 1]

12 Jan 2024

Understanding the current acute aortic syndrome (AAS) pathways - the Collaborative Acute Aortic Syndrome Project (CAASP) protocol

PONE-D-23-11061R1

Dear Dr. Zhong,

We’re pleased to inform you that your manuscript has been judged scientifically suitable for publication and will be formally accepted for publication once it meets all outstanding technical requirements.

Kind regards,

Academic Editor

PLOS ONE

Additional Editor Comments (optional):

Reviewers' comments:

Reviewer's Responses to Questions

**Comments to the Author**

1. Does the manuscript provide a valid rationale for the proposed study, with clearly identified and justified research questions?

Reviewer #2: Yes

2. Is the protocol technically sound and planned in a manner that will lead to a meaningful outcome and allow testing the stated hypotheses?

Reviewer #2: Yes

3. Is the methodology feasible and described in sufficient detail to allow the work to be replicable?

Reviewer #2: Yes

4. Have the authors described where all data underlying the findings will be made available when the study is complete?

Reviewer #2: Yes

5. Is the manuscript presented in an intelligible fashion and written in standard English?

Reviewer #2: Yes

6. Review Comments to the Author

You may also provide optional suggestions and comments to authors that they might find helpful in planning their study.

Reviewer #2: The authors have addressed all the queries satisfactorily.

There are no further comments.

Congratulations on the article.

7. PLOS authors have the option to publish the peer review history of their article (what does this mean?). If published, this will include your full peer review and any attached files.

Reviewer #2: **Yes: **Mohammed Idhrees

---

## [Editor Report · Acceptance letter]

25 Jan 2024

PONE-D-23-11061R1 

PLOS ONE

Dear Dr. Zhong, 

I'm pleased to inform you that your manuscript has been deemed suitable for publication in PLOS ONE. Congratulations! Your manuscript is now being handed over to our production team.

Kind regards, 

on behalf of

Dr. Robert Jeenchen Chen 

Academic Editor

PLOS ONE